# Evolution of thiolate-stabilized Ag nanoclusters from Ag-thiolate cluster intermediates

Yitao Cao[1,2], Jiahao Guo[1,2], Run Shi[1,2], Geoffrey I.N. Waterhouse[3], Jinheng Pan[4], Zhenxia Du[4], Qiaofeng Yao[5], Li-Zhu Wu[1], Chen-Ho Tung[1], Jianping Xie [5] & Tierui Zhang [1,2]

The synthesis of atomically precise thiolate-stabilized silver (Ag) nanoclusters is the subject of intense research interest, yet the formation mechanism of such nanoclusters remains obscure. Here, electrospray ionization mass spectrometry is successfully applied to monitor the reaction intermediates formed during the sodium-borohydride-reduction of silver 4-*tert*-butylbenzenethiolate (AgSPh-*t*Bu). We demonstrate a unique evolution route to thiolate-stabilized Ag nanoclusters mediated by Ag-thiolate clusters. The Ag-thiolate clusters form in the initial stage of reduction contain tens of Ag atoms and similar number of ligands, and they are transformed into $Ag_{17}(SPh-tBu)_{12}^{3-}$ and $Ag_{44}(SPh-tBu)_{30}^{4-}$ nanoclusters in the later reduction process. The number of Ag atoms in the Ag-thiolate clusters determines the reaction path to each final nanocluster product. A similar mechanism is found when silver 2,4-dimethylbenzenethiolate (AgSPhMe$_2$) is used as precursor. This mechanism differs markedly from the long-established bottom-up evolution process, providing valuable new insights into the synthesis of metal nanoclusters.

[1] Key Laboratory of Photochemical Conversion and Optoelectronic Materials, Technical Institute of Physics and Chemistry, Chinese Academy of Sciences, Beijing 100190, China. [2] University of Chinese Academy of Sciences, Beijing 100049, China. [3] School of Chemical Sciences, The University of Auckland, Auckland 1142, New Zealand. [4] College of Science, Beijing Key Laboratory of Environmentally Harmful Chemical Analysis, Beijing University of Chemical Technology, Beijing 100029, China. [5] Department of Chemical and Biomolecular Engineering, National University of Singapore, Singapore 119260, Singapore. Correspondence and requests for materials should be addressed to T.Z. (email: tierui@mail.ipc.ac.cn)

The synthesis of atomically precise metal nanoclusters has attracted tremendous research interest in recent years due to their unique physicochemical properties[1–4] which lead to end applications in catalysis[2,5,6], bioimaging[7,8], sensors[9,10], and many other areas. Amongst atomically precise metal nanoclusters, thiolate-stabilized gold (Au) and silver (Ag) nanoclusters have been the most widely investigated[5,11–16]. Although Au and Ag have similar atom radii and share a face-centered cubic (*fcc*) structure in crystalline materials, these metals have distinct differences in the types of thiolate-stabilized nanoclusters they form. For a single aliphatic thiol, atomically precise Au nanoclusters with a wide range of sizes can be synthesized, including $Au_{19}$[17], $Au_{20}$[18], $Au_{24}$[19], $Au_{25}$[20,21], $Au_{38}$[22,23], $Au_{68}$[24], and $Au_{144}$[25]. In contrast, for thiolate-stabilized Ag nanoclusters there is a far more rigid relationship between thiolate ligands and nanocluster sizes[26,27]. For each aromatic thiol, generally only a single atomically precise Ag nanocluster is formed[26,28–30].

The evolution process of thiolate-stabilized $Au_{25}$ nanoclusters from Au-thiolate precursors was recently studied by Xie et al. and a stepwise bottom-up formation process involving Au(I)-SR monomers and oligomers was developed[31]. However, the formation mechanism of atomically precise thiolate-stabilized Ag nanoclusters from Ag-thiolate precursors is not known, motivating a detailed investigation.

Herein, we demonstrate the successful identification of intermediates in the evolution process of atomically precise thiolate-stabilized Ag nanoclusters by electrospray ionization mass spectrometry (ESI-MS), thereby allowing the establishment of a unique evolution mechanism. As shown in Fig. 1, instead of forming monomers or oligomers followed by bottom-up evolution process as reported for Au nanoclusters[31], the reduction of the Ag-thiolate complex precursor, silver 4-*tert*-butylbenzenethiolate (AgSPh-*t*Bu), generates discrete Ag-thiolate clusters containing tens of Ag atoms and similar number of thiolate ligands (i.e., a similar Ag:thiolate composition to the Ag-thiolate precursor). It's worth mentioning that the Ag-thiolate precursor can be regarded as Ag(I)-thiolate complex with an infinite chain length and takes the form of insoluble light yellow flakes[30]. In contrast, the intermediate clusters can be regarded as a Ag-thiolate complex with a finite chain length. The intermediate clusters are subsequently transformed to thiolate-stabilized $Ag_{17}(SPh\text{-}t\text{Bu})_{12}^{3-}$ and $Ag_{44}(SPh\text{-}t\text{Bu})_{30}^{4-}$ nanoclusters through ligand dissociation and size focusing as the reduction proceeded. Further analysis of Ag atom numbers contained in the intermediate clusters reveals that the number of Ag atoms determines the reaction path to the final products. It should be mentioned that in a previous report, the bulky ligand SPh-*t*Bu was found to stabilize only $Ag_{17}(SPh\text{-}t\text{Bu})_{12}^{3-}$ rather than both $Ag_{17}(SPh\text{-}t\text{Bu})_{12}^{3-}$ and $Ag_{44}(SPh\text{-}t\text{Bu})_{30}^{4-}$, as is reported in the present study[28]. Our findings indicate that the size (number of Ag atoms) in the intermediate Ag-thiolate clusters plays a critical role in the reaction routes to the different thiolate-stabilized nanoclusters with the same capping ligand.

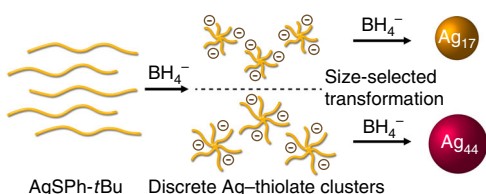

**Fig. 1** Schematic illustration of the evolution process. Thiolate-stabilized Ag nanoclusters were obtained through sodium-borohydride-reduction of AgSPh-*t*Bu precursor. Discrete Ag-thiolate clusters were captured by ESI-MS as the key intermediates in the evolution process

## Results

**Synthesis of Ag nanoclusters.** In a typical reaction, 20 mg of $AgNO_3$ (0.12 mmol) was dissolved in 10 mL of acetonitrile, and the resulting solution was added dropwise into a solution of 100 μL of HSPh-*t*Bu (0.60 mmol) in 20 mL of acetonitrile under vigorous stirring. A light yellow AgSPh-tBu precipitate was formed immediately. The suspension was kept for several minutes before 10 mg of $NaBH_4$ (0.26 mmol) powder was added into the reaction mixture under vigorous stirring. When the solution turned clear and colorless, 10 mL of methanol was added to promote the reduction by providing active hydrogen. Thiolate-stabilized Ag nanoclusters were obtained within 15 min at room temperature. UV–Vis absorption spectroscopy (Fig. 2b) identified characteristic absorption peaks of thiolate-stabilized $Ag_{44}$ species[14,27], while ESI-MS results confirmed the formation of two different stable products: $Ag_{17}(SPh\text{-}t\text{Bu})_{12}^{3-}$ and $Ag_{44}(SPh\text{-}t\text{Bu})_{30}^{4-}$ (Fig. 2a and Supplementary Fig. 1).

**Capture of reaction intermediates by ESI-MS.** Attention was subsequently focused on the evolution of thiolate-stabilized Ag nanoclusters during the reduction of the Ag-thiolate complexes. Preliminary attempts to identify the reaction intermediates under the original reaction conditions by ESI-MS were unsuccessful due to the high temperature in the ESI-MS instrument (source temperature: 40 °C, desolvation temperature: 120 °C), which caused the reaction to proceed to completion very quickly. Some small peaks for the intermediates could be detected (Supplementary Fig. 2), though it was impossible to systematically follow the evolution and disappearance of the intermediates due to the fast reaction kinetics. To suppress the reduction kinetics of the system, we replaced the methanol (10 mL) by 5 mL of methanol containing 2.5 mg of NaOH. In the reaction without NaOH, the pH of the reaction mixture was measured to be 9.3, while by adding NaOH, the solution pH was adjusted to above 14 (out of range of the pH meter). Under such new reaction conditions, the intermediates formed from the AgSPh-*t*Bu precipitate could be successfully captured by ESI-MS and they showed perfect coincidence with the small peaks captured under the former reaction conditions (Supplementary Fig. 3). However, it is worth mentioning that the final products, $Ag_{17}(SPh\text{-}t\text{Bu})_{12}^{3-}$ and $Ag_{44}(SPh\text{-}t\text{Bu})_{30}^{4-}$ were less stable under these new reaction conditions

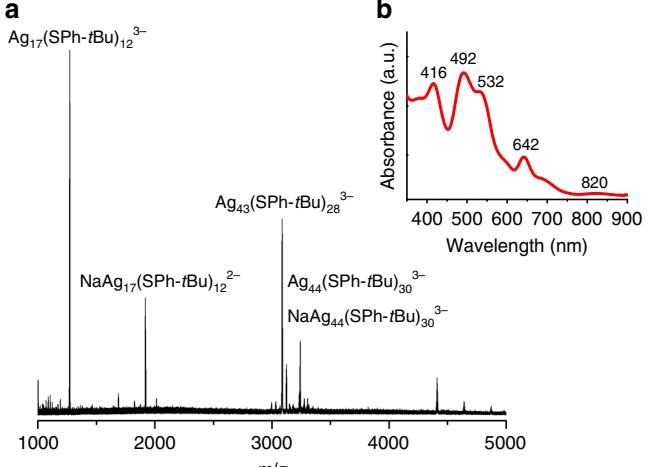

**Fig. 2** Characterizations of final products. **a** Negative-mode ESI-MS and **b** UV–Vis absorption spectrum of the final reaction product. Species with *m/z* larger than 4000 can also be assigned to $Ag_{44}(SPh\text{-}t\text{Bu})_{30}^{4-}$ and its fragments. See Supplementary Fig. 1 for more detailed analysis

since their evolution was accompanied by slight etching, as evidenced by the appearance of some miscellaneous small peaks in Fig. 3g.

**Analysis of ESI-MS spectra**. The entire evolution process of the $Ag_{17}(SPh-tBu)_{12}{}^{3-}$ and $Ag_{44}(SPh-tBu)_{30}{}^{4-}$ products is captured in the negative-mode ESI-MS spectra shown in Fig. 3. If the evolution of these products occurred by a stepwise bottom-up process (typical for Au nanoclusters)[31], then lower molecular weight intermediates should be consumed with time. However, all low molecular weight species ($m/z < 1000$) identified in Fig. 3a, d can be assigned to inorganic ions, such as $Na^+$ and $NO_3{}^-$, in the reaction mixture (Supplementary Fig. 4). Instead, a series of peaks with a $-3$ charge and a characteristic Ag isotopic pattern was observed in the $m/z$ 2000–5000 region. A straightforward relationship is apparent between these species and the two final products $Ag_{17}(SPh-tBu)_{12}{}^{3-}$ and $Ag_{44}(SPh-tBu)_{30}{}^{4-}$. As the reaction proceeded, the signals for the two final products intensified whilst signals for the species formed early in the reaction were attenuated, leading to a conclusion that these species are key reaction intermediates. $Ag_{17}(SPh-tBu)_{12}{}^{3-}$ and $Ag_{44}(SPh-tBu)_{30}{}^{4-}$ were also found to evolve with different speeds, suggesting two separate routes to these products. As shown in Fig. 3, $Ag_{17}(SPh-tBu)_{12}{}^{3-}$ (Fig. 3a, d) evolved more quickly than $Ag_{44}(SPh-tBu)_{30}{}^{4-}$ (Fig. 3c, g). Accordingly, we were able to distinguish two sets of intermediates, one for each final product. Note that attempts were also made to detect positively charged or neutral intermediates during the reaction. Except for species associated with inorganic ions in the low molecular weight range, no obvious signals corresponding to reaction intermediates could be found (Supplementary Fig. 5).

The intermediate clusters leading to $Ag_{17}(SPh-tBu)_{12}{}^{3-}$ have $m/z$ 2000–2800 and become detectable almost immediately after the reaction initiated. Detailed analyses of peaks in Fig. 3a are shown in Supplementary Figs. 6-9. Selected peak positions and their assignments are listed in Supplementary Table 1. These intermediates, denoted here as intermediate Ag-thiolate clusters, have a general formula $Ag_n(SPh-tBu)_{n-x}$, where $0 \leq x \leq 4$. The number of Ag atoms in these intermediate clusters falls in the range of 22–34. They have a Ag:thiolate ratio similar to the Ag-thiolate complex precursor (Ag:thiolate = 1:1) and they are negatively charged due to the reduction by $NaBH_4$. Intermediate Ag-thiolate clusters leading to the formation of $Ag_{44}(SPh-tBu)_{30}{}^{4-}$ have $m/z$ 2800–5000, and contain 35–57 Ag atoms (Fig. 3c, Supplementary Figs. 10, 11 and Supplementary Table 2). Thus, the number of Ag atoms in the intermediate Ag-thiolate clusters has a strong influence on the final products. In this case, Ag-thiolate clusters containing 22–34 Ag atoms will be transformed to $Ag_{17}(SPh-tBu)_{12}{}^{3-}$, whilst those with 35–57 Ag atoms will lead to the formation of $Ag_{44}(SPh-tBu)_{30}{}^{4-}$. The initially formed Ag-thiolate clusters with polydisperse size (different Ag atom number in clusters) will 'size-focus' to their corresponding stable atomically precise thiolate-stabilized Ag nanoclusters later in the reaction. This process is also accompanied by reduction of Ag(I) to Ag(0), which will enhance the metal–metal interaction between Ag atoms in the clusters and result in simultaneous thiolate ligand dissociation.

**Evolution process of Ag nanoclusters**. To allow better understanding of the transformation from the intermediate Ag-thiolate clusters to the final products (atomically precise thiolate-stabilized Ag nanoclusters), we defined the value $M = n - m + l$ to represent the degree of reduction of each intermediate Ag-thiolate cluster (also known as the valence electron count in nanoclusters)[32], where $n$ is the number of Ag atoms, $m$ is the number of thiolate ligands and $l$ is the number of units of negative charge on the cluster, respectively. As described in Supplementary Fig. 12, for the Ag-thiolate complex precursor (before the reducing agent was added), $M = 0$ (since $n = m$, $l = 0$). When the precursor was dissolved and transformed to the intermediate Ag-thiolate clusters, $M$ increased due to the reduction. For the intermediate Ag-thiolate clusters, calculation of $M$ was based on the general formula $Ag_n(SPh-tBu)_{n-x}$, where $0 \leq x \leq 4$. $M$ falls in the range of 3 to 7. At this stage of reaction, the intermediate Ag-thiolate clusters contained a large amount of Ag(I) ions and showed similar composition to the Ag-thiolate complex (intermediate Ag-thiolate clusters can be regarded as slightly reduced Ag-thiolate complex with a limited chain length, in which some of the Ag(I) were reduced to Ag(0) to give negatively charged clusters). Accordingly, we denote the intermediates as intermediate Ag-thiolate clusters. In a system with a strong reducing power ($NaBH_4$ and methanol), the Ag-thiolate clusters with relatively small $M$ were unstable and were further reduced to species with larger $M$, e.g., $Ag_{17}$ ($Ag_{17}(SPh-tBu)_{12}{}^{3-}$, $M = 8$) and $Ag_{44}$ ($Ag_{44}(SPh-tBu)_{30}{}^{4-}$, $M = 18$). Note that we could not identify any species with $M$ in the range of 8 to 18. We hypothesized that the intermediate species with $M$ larger than 8 were highly reactive, and they were transformed to $Ag_{44}(SPh-tBu)_{30}{}^{4-}$ before they could be captured by ESI-MS. This can be rationalized by a shell closing phenomenon wherein clusters with $M = 8, 18, 34$, etc have higher stability than the others[32].

**TEM observation of intermediates**. Transmission electron microscopy (TEM) was applied to monitor the intermediates at the very beginning of the reduction reaction. After addition of $NaBH_4$ to the reaction mixture containing the AgSPh-tBu precipitate, the yellow turbid solution turned clear and colorless. A sample was taken and subjected to TEM analysis. Figure 4 shows a representative TEM image of the sample, revealing nanoparticles with a mean size of $1.24 \pm 0.18$ nm. $Ag_{44}$, the main product in our reaction system, has been reported to have a core

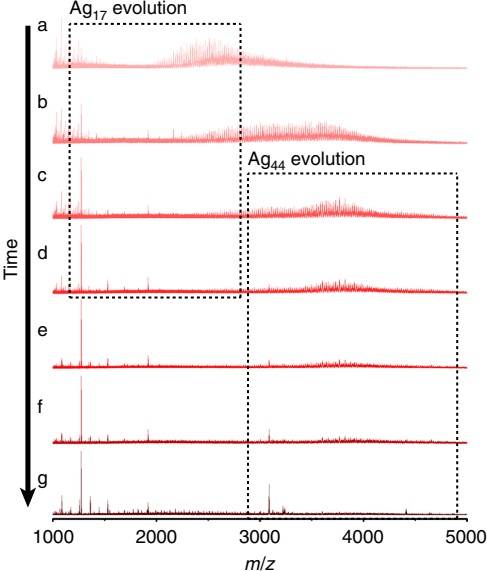

**Fig. 3** Time-dependent ESI-MS spectra. The spectra show the evolution of atomically precise thiolate-stabilized Ag nanoclusters in the reaction solution. The rectangles highlight the evolution of the final products, $Ag_{17}(SPh-tBu)_{12}{}^{3-}$ and $Ag_{44}(SPh-tBu)_{30}{}^{4-}$. The capture started as soon as the methanol was added and the reaction time in each panel was **a** 0 min, **b** 2 min, **c** 4 min, **d** 7 min, **e** 10 min, **f** 15 min and **g** 25 min

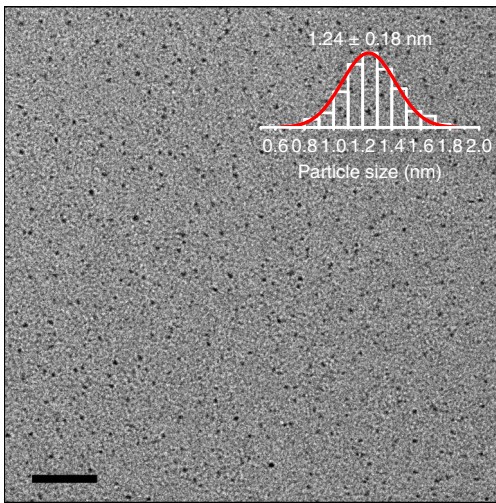

**Fig. 4** Particle size of intermediates. TEM image of the intermediates obtained after NaBH$_4$ addition to the reaction mixture, when the mixture became clear and colorless. Scale bar: 20 nm. The overlaid inset shows a particle size histogram and the corresponding Gaussian fit

diameter of ~1.2 nm[14,27,32]. The perfect coincidence in size lends supports to the earlier conclusion from the ESI-MS data that the discrete nanoparticles formed in this stage (i.e., intermediate Ag-thiolate clusters) are the intermediates in the formation of Ag$_{17}$(SPh-$t$Bu)$_{12}$$^{3-}$ and Ag$_{44}$(SPh-$t$Bu)$_{30}$$^{4-}$.

**Control experiment through a bottom-up process**. The final products of this reaction, Ag$_{17}$(SPh-$t$Bu)$_{12}$$^{3-}$ and Ag$_{44}$(SPh-$t$Bu)$_{30}$$^{4-}$, were not particularly stable, as evidenced by the color fading of the reaction mixture (to colorless) over several hours if the solution was kept in air at room temperature under constant stirring (please refer to Supplementary Note 1 for more details concerning the stability of Ag$_{44}$ and Ag$_{17}$). ESI-MS showed that the colorless species in solution were Ag-thiolate monomers and oligomers, amongst which Ag(SPh-$t$Bu)$_2$$^-$ and Ag$_5$(SPh-$t$Bu)$_6$$^-$ had the highest abundance (Supplementary Fig. 13). This mixture of Ag-thiolate monomers and oligomers served as a perfect precursor to investigate the possibility of synthesizing Ag$_{17}$(SPh-$t$Bu)$_{12}$$^{3-}$ and Ag$_{44}$(SPh-$t$Bu)$_{30}$$^{4-}$ via a bottom-up process simply by adding a small amount of NaBH$_4$ to induce a reduction reaction. However, upon addition of 10 mg of NaBH$_4$ to the colorless mixture, none of the absorption features of thiolate-stabilized Ag nanoclusters were observed. Instead, a broad peak corresponding to the localized surface plasmon resonance (SPR) of large Ag nanoparticles (typically with particle size above 3 nm) was observed (Supplementary Fig. 14a). This data were further verified by TEM measurements, where irregular Ag nanospheres of particle size larger than 3 nm were clearly seen in the representative TEM images (Supplementary Fig. 14b). This experiment demonstrates conclusively that it is not possible to obtain thiolate-stabilized Ag nanoclusters such as Ag$_{17}$(SPh-$t$Bu)$_{12}$$^{3-}$ and Ag$_{44}$(SPh-$t$Bu)$_{30}$$^{4-}$ via a bottom-up process under the experimental conditions used in this study.

**Experiments using 2,4-dimethylbenzenethiol (HSPhMe$_2$)**. To test the validity of the aforementioned Ag nanocluster evolution process, an analogous set of experiments were performed using a different thiolate ligand, HSPhMe$_2$. HSPhMe$_2$ has been reported to stabilize Ag$_{25}$ nanoclusters, thus it was of interest to see if nanoclusters of this particular size could be formed through Ag-thiolate cluster intermediates[26]. ESI-MS experiments showed that at the early stage of the reaction, the intermediate Ag-thiolate clusters

were formed (detected by a series of products with −3 charge), which in turn served as key intermediates during the evolution process of thiolate-stabilized Ag nanoclusters (Supplementary Figs. 15, 16). Multiple thiolate-stabilized Ag nanoclusters were detected as products (Supplementary Fig. 17). Interestingly, the main final products were Ag$_{26}$(SPhMe$_2$)$_{19}$$^{2-}$ and Ag$_{44}$(SPhMe$_2$)$_{30}$$^{4-}$, but no signals corresponding to Ag$_{25}$(SPhMe$_2$)$_{18}$$^-$ were detected[26]. The two products Ag$_{26}$(SPhMe$_2$)$_{19}$$^{2-}$ and Ag$_{44}$(SPhMe$_2$)$_{30}$$^{4-}$ evolved simultaneously, thus it was difficult to identify the intermediate Ag-thiolate clusters leading to each product. However, the overall evolution of the Ag$_{26}$(SPhMe$_2$)$_{19}$$^{2-}$ and Ag$_{44}$(SPhMe$_2$)$_{30}$$^{4-}$ products was similar to that presented in Fig. 2 for Ag$_{17}$(SPh-$t$Bu)$_{12}$$^{3-}$ and Ag$_{44}$(SPh-$t$Bu)$_{30}$$^{4-}$, which provides a sufficient evidence to suggest a similar evolution process. Ag$_{17}$ nanoclusters were also detected during the evolution process of Ag$_{26}$(SPhMe$_2$)$_{19}$$^{2-}$ and Ag$_{44}$(SPhMe$_2$)$_{30}$$^{4-}$, and they were disappeared as the reaction proceeded (Supplementary Figs. 15 and 18). Considering the different ESI-MS fragment patterns of Ag$_{44}$ capped by SPhMe$_2$ (Supplementary Fig. 17c and 17d) and Ag$_{44}$ capped by SPh-$t$Bu (Fig. 2a and Supplementary Fig. 1), we conclude that Ag$_{17}$ and Ag$_{44}$ were formed in the presence of each thiolate capping ligand, though their stabilities differ. Furthermore, some capping ligands can generate unique thiolate-stabilized Ag nanoclusters, such as Ag$_{26}$(SPhMe$_2$)$_{19}$$^{2-}$.

## Discussion

Previous studies have shown atomically precise Ag$_{44}$ nanoclusters are inert and do not "seed" the synthesis of larger thiolate-stabilized Ag nanoclusters, whereas Au$_{25}$ nanoclusters could seed larger nanoclusters when the analogous reaction was performed[14]. Our results offer a highly plausible explanation for the non-seeding properties of Ag nanoclusters, since the evolution of Ag$_{44}$ nanoclusters is based on a process from discrete Ag-thiolate cluster intermediates, a pathway that prevents the growth of larger nanoclusters. Conversely, Au$_{25}$ nanoclusters are formed through a stepwise bottom-up process[31], providing the distinct possibility for Au$_{25}$ to react with other Au-thiolate monomers to create larger nanoclusters.

Recently, thiolate-stabilized Ag nanoparticles containing 136 and 374 Ag atoms have been successfully synthesized[33], demonstrating potential for the synthesis of larger thiolate-stabilized Ag nanoclusters. Results of this study demonstrate that the final size of the Ag nanoclusters is determined by the size of intermediate Ag-thiolate clusters. Accordingly, if larger intermediate Ag-thiolate clusters could be generated during the reaction, then it should be possible to obtain larger Ag nanoclusters at the end of the reaction. However, all the Ag-thiolate clusters we detected by ESI-MS contained less than 60 Ag atoms. Structure of the AgSPh-$t$Bu complex precursor (the initial yellow precipitate obtained before NaBH$_4$ addition) may help to understand this limitation. As shown in Supplementary Fig. 19, the AgSPh-$t$Bu complex consisted of particles with size of only 1–2 nm, very similar to that of the intermediate Ag-thiolate clusters (Fig. 4). These data suggest that the size of the assembled particles in the AgSPh-$t$Bu precursor complex may directly relate to the number of Ag numbers in the intermediate Ag-thiolate clusters.

To summarize, we demonstrate that the synthesis of atomically precise thiolate-stabilized Ag nanoclusters by chemical reduction of Ag-thiolate complex precursors proceeds via an evolution process involving Ag-thiolate cluster intermediates. These intermediate clusters contain tens of Ag atoms and a similar number of thiolate ligands, and they were transformed to Ag nanoclusters of well-defined size through thiolate dissociation and size-focusing as the reduction reaction proceeds. Furthermore, the

number of Ag atoms in the intermediate Ag-thiolate clusters determines the reaction pathway and final size of the thiolate-stabilized Ag nanoclusters. Our findings significantly advance the understanding of the formation mechanism of atomically precise Ag nanoclusters, and hint at efficient routes towards Ag nanoclusters of different (e.g., larger) sizes.

## Methods

**Chemicals**. 4-*tert*-butylbenzenethiol (HSPh-*t*Bu) and 2,4-dimethylbenzenethiol (HSPhMe$_2$) were purchased from Aladdin Chemistry Co. Ltd. Sodium borohydride (NaBH$_4$, 99.99%, metal basis) was purchased from Sigma-Aldrich. Methanol, acetonitrile, and NaOH were obtained from local suppliers and used without further purification.

**Synthesis of thiolate-stabilized Ag nanoclusters**. In a typical reaction, 20 mg of AgNO$_3$ (0.12 mmol) was dissolved in 10 mL of acetonitrile, and the resulting solution was added dropwise into a solution of 100 µL of HSPh-*t*Bu (0.60 mmol) in 20 mL of acetonitrile under vigorous stirring. A light yellow AgSPh-*t*Bu precipitate was formed. 10 mg of NaBH$_4$ (0.26 mmol) powder was added into the reaction mixture under vigorous stirring, and the stirring continued for several minutes. When the solution turned clear and colorless, 10 mL of methanol was added to promote the reduction. Thiolate-stabilized Ag nanoclusters were obtained within 15 min at room temperature.

**Capture of intermediate species**. In the above synthesis, the 10 mL methanol was replaced with 5 mL of methanol solution containing 2.5 mg of NaOH. All other conditions were unchanged.

**UV–visible absorption spectroscopy**. Solution-phase absorption spectra were recorded using standard 1 cm path length quartz cuvettes on a Hitachi UV-3900 spectrophotometer.

**Transmission electron microscope (TEM)**. TEM images were obtained using a JEOL-2100F microscope operating at an accelerating voltage of 200 kV. Samples were dispersed on hydrophilic carbon films for analysis.

**Electrospray ionization mass spectrometry (ESI-MS)**. All mass spectrometry data were collected on a Xevo G2-S quadrupole time-of-flight mass spectrometer (Waters, Milford, USA) equipped with an ESI source and operated in both positive and negative ionization mode. Instrumental parameters were as follows: capillary voltage: 2.5 kV, sample cone voltage: 15 V, extraction cone voltage: 3.0 V, collision energy: 0 eV, cone gas flow rate: 30 L/h, desolvation gas flow rate: 800 L/h, source temperature: 40 °C, and desolvation temperature: 120 °C. The scan range was from 50 to 7000 Da. Data acquisition, handling, and instrument control were performed by using UNIFI.

**Data availability**. All relevant data are available from the corresponding author on request.

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

## Acknowledgements

T.Z. is grateful for financial support from the National Key Projects for Fundamental Research and Development of China (2017YFA0206904, 2017YFA0206900, 2016YFB0600901), the Ministry of Science and Technology of China (2014CB239402), the National Natural Science Foundation of China (51772305, 51572270, U1662118), the Strategic Priority Research Program of the Chinese Academy of Sciences (XDB17000000), the Royal Society-Newton Advanced Fellowship (NA170422), the International Partnership Program of Chinese Academy of Sciences (GJHZ1819), the Young Elite Scientist Sponsorship Program by CAST (YESS), and the Youth Innovation Promotion Association of the CAS.

## Author contributions

T.Z. and Y.C. conceived the idea and design the experiments. T.Z. supervised the project. Y.C. carried out the experiments and characterizations. Y.C., J.P., and Z.D. performed the ESI-MS tests. Y.C., G.I.N.W., and T.Z. wrote the manuscript. J.G., R.S., Q.Y., L.W., C.T., and J.X. discussed the results and commented on the manuscript.

## Additional information

**Competing interests:** The authors declare no competing interests.

