## [Peer Review File · Nature Communications]

Reviewers' comments:

Reviewer #1 (Remarks to the Author):

Ligand protected metal clusters have attracted much attention due to have their unique properties which do not show in correspond bulk metal. In this field, since many works concerning thiolate protected gold clusters have been reported, much knowledge about has been accumulated. Recently, in addition to these gold clusters, thiolate protected silver clusters has also been studied. Although many novel silver clusters have been found after being reported precise synthesis of silver cluster, their information is less than that of gold clusters. In this work, the authors explored the intermediate clusters which occur during synthesis of silver clusters, and investigated the mechanism of synthesis of silver clusters. Though the work about mechanism of thiolate protected gold clusters has already conducted, I have never seen about silver cluster case. Thus, I think the novelty of this work is high, this thesis deserves as communication. In this manner, I can permit publication of this thesis to Nature Communications. However, since I have some comments and questions, I would like you to consider about these at the revision.

(1) In the main text, silver thiolate cluster having 22-34 silver atoms transforms to Ag₁₇ cluster, and silver thiolate cluster having 35-37 silver atoms transforms to Ag₄₄ cluster. I think that authors investigated about this phenomenon well by ESI spectrometry. However, I would like to know why the difference of silver thiolate cluster affects final production. Additionally, In the case of Ag₁₇ cluster, etching process seems to be occurred, but Ag₄₄ cluster case, ripening process seems to be occurred. Do you have any reasons or opinions about this difference?

(2) In my opinion, although Ag₄₄ cluster is comparatively stable, this cluster was decomposed after reaction in this work. Why did this occur? Additionally, when authors added NaBH₄ to the samples after decomposing, cluster size increased. I would like to know the reason.

(3) As authors also say in main text, HSP_hMe₂ is used to synthesize Ag₂₅ cluster. However, in this experiment, main final products are Ag₂₆ and Ag₄₄ clusters. I could not understand why Ag₂₅ cluster could not synthesize.

(4) Overall, experimental is splendid, and phenomena are very interesting to me. But, as I mentioned above, there are not more description about reasons of these interesting phenomena very much. If authors will think about this point, this work will be better.

Reviewer #2 (Remarks to the Author):

This report claims to show that Ag nanoclusters are formed through a evolution process that involves intermediates containing tens of Ag atoms. Electrospray method was used to characterise the intermediates.

The ESI MS method shows only charged ions in the reported analysis conditions. Neutral species will not be detected.

Some of the authors previously reported (reference 31) a similar evolution study on Au₂₅(SR)₁₈. Judging both from citations and personal observation, the novelty and usefulness of ref. 31 (and the current report) is unclear.

The conclusions are only partly reliable because only negatively charged species are observed. Other literature references for electrospray method, especially Ag clusters, should be cited. The report claims to provide "valuable new insights". This referee does not see the interest of this work to others in the community.

Reviewer #3 (Remarks to the Author):

Summary of Key Points

ESI-MS was used to monitor reactive intermediates and discretely sized Ag nanocluster products over time using two bulky, aromatic thiols as capping agents. The ESI-MS spectra provided here does not show signals corresponding to low molecular weight Ag-thiolate monomers or oligomers. This indicates that the Ag₁₇ and Ag₄₄ product nanoclusters in this work are not formed through a "bottom-up" approach, which is commonly reported for Au clusters; this is an original, significant and surprising result. The reactive intermediates instead appear to be clusters themselves, which are similar in atom count to the final nanocluster product, and are likely "size-focused" to the observed discretely sized cluster products. Based on this, the authors propose relative amounts of ligand:metal:charge (denoted M) that predisposed each clusters formation based on shell closure magic numbers and suggest that larger Ag clusters could be formed if the correct "M ratio" is observed in the intermediate cluster.

Validity

There are no major flaws in the work presented here to indicate that it is unfit for publication.

Originality and Significance

The absence of oligomeric precursors in the formation of discrete Ag clusters is an original and significant result because it indicates that these Ag nanoclusters are formed via a different pathway than similarly sized Au clusters. The "M ratio" provided here could be useful to predict relative ratios of precursor chemicals to force the formation of larger clusters with discrete sizes offering some semblance of size control. This could be of broad scientific interest.

Data and Methods

In general, the science of the paper is sound, but there were a handful of places where I wanted a few more details to either lend support or to be fully convinced of their claims. For example, the cluster synthesis appears to be relatively standard and both UV-Vis and TEM supports the formation of the Ag₄₄ as seen in the literature. However, for the sake of reproducibility, they should indicate the amount of time it takes for the yellow precipitate precursor to form before reduction to clusters. Since this is also an 'intermediate' in the synthesis of their clusters, I'm curious as to its stability/shelf life and whether differently sized clusters are seen when this is aged for hours or days? Is there any ESI-MS data on this precursor precipitate to get an idea of how "finite" these precursor chains are (i.e. is it the same atom count as the proposed intermediate clusters?) This could lend support to their proposed mechanistic pathways.

The ESI-MS data clearly indicates the formation of discrete Ag₁₇ and Ag₄₄ product clusters over time. There is also a noted absence of smaller MW peaks that would be attributed to small Ag(I)-thiolate monomers or oligomers, which supports the assertion that a bottom up mechanism may not operate here. Instead the reactive intermediates observed with ESI-MS appear to have a very broad range that centers around the peak observed for each cluster product. The broad intermediate peaks decrease over time as the product cluster signals intensify, indicating that size-focusing must be occurring on both clusters as the reaction proceeds to completion. The authors note that Ag₁₇ could be formed from "intermediate clusters" of 22 – 34 atoms and Ag₄₄ from 35 – 57 atoms and provide a

formula based on the amounts of metal, ligand and charge to predict which cluster forms under which conditions. It appears both clusters do form in a unique way based on the identity and timing of ESI peaks in Figure 3. However, I am not entirely convinced that the “intermediate clusters” giving rise to the ~2800 m/z peaks aren't also being used as a scaffold for further reduction into the larger Ag44 cluster in addition to or instead of forming Ag17. How can they be sure these higher MW intermediates form only Ag17 and not also Ag44? Is there a ratio of intensity for the appearance/disappearance of the useful peaks to completely support this assertion? It appears to me that there are many peaks around the Ag17 cluster that disappear over time as the Ag17 peak becomes more intense, which could be the only intermediates for Ag17. And there is a huge gap in the ESI spectrum between the Ag17 peak and the purported intermediate clusters that give rise to Ag44. While for Ag44, it appears like an early ~2800 m/z intermediate that could undergo Ostwald ripening to form larger intermediate clusters that are later size focused into the major Ag44 product. A quantifiable metric would be of help to rule this possibility out.

Given that the kinetics of cluster formation were too rapid in ESI to be of practical use for detecting intermediates, they reduced the volume of methanol by half and added 50 mol % of NaOH to slow the reaction progress down. I presume this addition of hydroxide is to quench the active hydrogen, but the authors' do not explicitly address the role of hydroxide ion and how it is involved in the mechanism they propose. They do note that the products formed in the presence of hydroxide ion were less stable over time, and showed evidence of other un-explained fragments in ESI, so how can they be sure the same mechanism will operate in neutral conditions versus basic conditions? Was the pH explicitly monitored in both synthetic conditions? Also, did the addition of hydroxide cause any Ag(OH) to precipitate out? A gravimetric type of analysis here could indicate how much Ag is “tied up” in the precipitate cluster and how much is free in solution, which could be useful for a more accurate determination of its elemental composition. Additionally, if the hydroxide does in effect lower the amount of hydride in solution, did the authors try the experiment using less borohydride to see if the same ESI peaks are observed in each condition? In my experience, the amount and/or “freshness” of this reducing agent can noticeably change the size distribution of product clusters.

After a period of hours, ESI indicates the clusters had degraded into small monomers and oligomers (typical as preliminary intermediates for Au clusters). These were used as a control to see if a bottom up approach could work for the formation of the discrete clusters by adding NaBH₄. When they did this, AgNPs formed, but they were much larger than Ag17 and Ag44, which supports that maybe a bottom up mechanism doesn't work for these small ‘electronically stabilized’ clusters, but perhaps could work for the larger ‘geometrically stabilized’ clusters. However, it seems that the authors added half as much borohydride for this control reaction than for the initial reaction – there is no indication of the sample volume in this control reaction, so these may be consistent molar ratios with the earlier experiment, but without more details it's hard to know for sure. As stated earlier, the relative amount of borohydride used to generate noble metal nanoparticles has an effect on the size of the clusters generated, so this should be addressed to ensure it's a proper control. This is analogous to the earlier point, where the amount of “active” borohydride should be less with the addition of hydroxide ion. Does this change in the amount of reducing agent affect the size of these clusters?

Conclusions

It's reasonable to conclude that the Ag17 clusters synthesized in this work are generated from other similarly sized clusters through size-focusing events that occur shortly after reduction occurs, instead of from smaller oligomeric units that “build up” to the magic number of atoms for shell closure. It is possible the same type of phenomenon is occurring for Ag44, but without a quantifiable metric of relative peak intensities, I'm not entirely convinced that Ag44 isn't formed through smaller clusters and later Ostwald ripening. Either way, it is a very interesting result/absence of result that no monomers or small oligomers of silver-thiolate complexes are observed as is seen with gold

nanoclusters. This does represent a new pathway for cluster formation and could potentially allow researchers to attempt size control of these clusters using specified ratios of chemical precursors.

Improvements

Generally, the document was written well, but there were a few things that confused me or that I thought could be improved to clarify and strengthen the paper.

- Adding those missing details or rationalizations stated above would support their assertions and leave less room for misinterpretation.
- The mechanistic description needs to be very clear. The word 'clusters' is used for both a product and an intermediate, and the words 'precursors' is used for two different types of precursor chains (infinite and finite). It's easy to get lost in this part of the text in particular. For example, it may be helpful for the reader if the authors explicitly use "intermediate clusters" or even an acronym like IC throughout the text to ensure clarity.
- Some things appear contradictory. For example, they comment on the remarkable stability of Ag17 and Ag44 clusters, but later go on to describe how these particular Ag17 and Ag44 clusters weren't particularly stable and decomposed within hours. Addressing this with an explanation (that also ties into the mechanism) would be appreciated.
- A little bit of organizational clean up could improve the readability. There are a few awkward sentences, and odd conclusions drawn at random points in the text that either don't follow the previous statement, belong elsewhere or are simply redundant.
 - o Line 23 spacing typo
 - o line 62 first sentence seems out of place/awkward with rest of text
 - o line 64 weird conclusion from previous statement
 - o lines 77-78 redundant from previous.
 - o Lines 88-90 awkward way to describe the reaction steps.
 - o 115-117 awkward
 - o 153 sentence about TEM needs context.
 - o Line 153 spacing typo
 - o Line 201 missing "of"?
 - o Lines 220-221 "stable thiolate-stabilized" is redundant

Reply to Reviewers' Comments

Comments from Reviewer #1:

General Comments: *Ligand protected metal clusters have attracted much attention due to have their unique properties which do not show in correspond bulk metal. In this field, since many works concerning thiolate protected gold clusters have been reported, much knowledge about has been accumulated. Recently, in addition to these gold clusters, thiolate protected silver clusters has also been studied. Although many novel silver clusters have been found after being reported precise synthesis of silver cluster, their information is less than that of gold clusters. In this work, the authors explored the intermediate clusters which occur during synthesis of silver clusters, and investigated the mechanism of synthesis of silver clusters. Though the work about mechanism of thiolate protected gold clusters has already conducted, I have never seen about silver cluster case. Thus, I think the novelty of this work is high, this thesis deserves as communication. In this manner, I can permit publication of this thesis to Nature Communications. However, since I have some comments and questions, I would like you to consider about these at the revision.*

Response: We thank the reviewer for his/her very positive comments.

Comment 1. *In the main text, silver thiolate cluster having 22-34 silver atoms transforms to Ag₁₇ cluster, and silver thiolate cluster having 35-57 silver atoms transforms to Ag₄₄ cluster. I think that authors investigated about this phenomenon well by ESI spectrometry. However, I would like to know why the difference of silver thiolate cluster affects final production. Additionally, In the case of Ag₁₇ cluster, etching process seems to be occurred, but Ag₄₄ cluster case, ripening process seems to be occurred. Do you have any reasons or opinions about this difference?*

Response: We rationalize this phenomenon in terms of the reduction process of Ag-thiolate clusters and the relative stability of Ag nanoclusters. According to shell closing phenomenon, metal nanoclusters containing 8, 18, 34, etc. close shell electrons will have higher stability than other clusters. The Ag₁₇ and Ag₄₄ nanoclusters contain 8 and 18 close shell electrons, respectively. This explains why they are more stable, and the good stability of such nanoclusters has also been confirmed in several previous reports (*J. Am. Chem. Soc.* **2015**, 137, 11550; *Nature* **2013**, 501, 399; *Nat. Commun.* **2013**, 4, 2422).

Based on the above knowledge, we can draw a representative potential energy level for Ag nanoclusters containing different Ag atom numbers, and the Ag₁₇ and Ag₄₄

nanoclusters should represent the two minima, as shown in Fig. R1. In our first reaction (AgSPh-*t*Bu as the precursor), as the reduction proceeds, the potential energy of the intermediate Ag-thiolate clusters decreases. In this process, the Ag-thiolate clusters and the further reduced species with Ag atom numbers in different ranges will fall into one of two ‘potential wells’ (i.e. Ag₁₇ and Ag₄₄) in a system with strong reducing capability. During the transformation process, the intermediate species with more Ag atoms than the final product will go through an etching process (leading to Ag₁₇), and those have a wide distribution will go through a ripening process (leading to Ag₄₄). In our opinion, they should all go through a reduction process to reach the potential energy minima (i.e. Ag₁₇ and Ag₄₄).

Figure R1. Representative potential energy levels for Ag nanoclusters with different Ag atom numbers capped by SPh-*t*Bu.

Following this explanation, we can also draw a representative curve when using AgSPhMe₂ as precursor based on our experimental results, as shown in Fig. R2.

Figure R2. Representative potential energy level for Ag nanoclusters with different Ag atom numbers capped by SPhMe₂.

It is worth mentioning that the real potential energy level diagrams for these systems will be far more complicated than those mentioned above, as it must contain the full description of the whole reduction processes, not only several final states. 3D

potential energy level diagram are required, which are unlikely to be as smooth as the 2D curves described above. They will likely contain some shallow ‘potential wells’ that represent metastable species forming under some special reaction conditions. Further, they will need to include species with larger Ag atom numbers, leading to larger stable Ag nanoclusters.

Comment 2. In my opinion, although Ag₄₄ cluster is comparatively stable, this cluster was decomposed after reaction in this work. Why did this occur? Additionally, when authors added NaBH₄ to the samples after decomposing, cluster size increased. I would like to know the reason.

Response: Actually, several published reports on Ag₄₄ nanoclusters have suggested that Ag₄₄ is prone to oxidation at room temperature, with full oxidation occurring in a matter of hours (*Angew. Chem.*, **2009**, *121*, 6035; *Nanoscale*, **2012**, *4*, 4269; *J. Am. Chem. Soc.*, **2012**, *134*, 11856). This was also observed in our experiment. However, Ag₄₄ does show higher stability when it was capped by some special thiolate ligands, such as 5-mercapto-2-nitrobenzoic acid (*J. Mater. Chem. A*, **2013**, *1*, 10148). The nature of the capping thiolate thus appears important to Ag nanocluster stability, and may help explain the ‘ultrastable’ property of Ag₄₄ reported in *Nature* **2013**, *501*, 399, in which Ag₄₄ is capped by 4-mercaptobenzoic acid.

On the other hand, temperature and solvent might also play an important role in the stability of Ag₄₄ nanoclusters. If we kept our final product in a freezer at -18 °C instead of room temperature, the final products were stable for several days. If we diluted the reaction mixture with extra acetonitrile, the etching process was completed in 10 min at room temperature. An additional note concerning the stability of Ag₄₄ was provided in Supplementary Note 1.

After decomposing the Ag nanoclusters, they could be completely transformed to Ag-thiolate monomers and oligomers (Supplementary Fig. 12). When we added NaBH₄ to this solution, larger Ag nanoparticles were formed. This observation has direct relationship with the reaction route. When the precursor consists of Ag-thiolate monomers and oligomers, it is a straightforward conclusion that the reduction process leading to the formation of Ag (0) species will adopt a bottom-up route. This evolution route is absolutely different from that reported in the present study, where the evolution of Ag nanoclusters is mediated by Ag-thiolate cluster intermediates. As described in the Discussion section in the revised manuscript, the reduction process from the discrete Ag-thiolate cluster intermediates will prevent initially formed small clusters ‘seeding’ the growth of larger particles. On the contrary, in the reduction

process of monomers and oligomers, there is a distinct possibility for the formed clusters will combine further with monomers to create larger particles.

Comment 3. As authors also say in main text, HSPhMe₂ is used to synthesize Ag₂₅ cluster. However, in this experiment, main final products are Ag₂₆ and Ag₄₄ clusters. I could not understand why Ag₂₅ cluster could not synthesize.

Response: A very good point. We also noticed this phenomenon in our work. These two species have similar formulas. The formula of Ag₂₆ determined by ESI-MS is Ag₂₆(SPhMe₂)₁₉²⁻, and the formula of Ag₂₅ reported is Ag₂₅(SPhMe₂)₁₈⁻. They only differ by an AgSPhMe₂⁻ fragment. Actually, from our preliminary data, we also found that they might have similar absorption characteristics. The absorption spectrum of Ag₂₆ and Ag₄₄ mixture we obtained in the experiment is shown in Fig. R3a. Compared to a representative adsorption spectrum of Ag₄₄ (Fig. R3b), a distinct absorption peak appeared at ~680 nm, indicating a characteristic absorption peak corresponding to Ag₂₆ species. Interestingly, Ag₂₅ also shows an absorption peak at 675 nm. Similar absorption characteristics may indicate some similarities in their molecular structures. Considering the close shell electron number of Ag₂₆ is 9, while that of Ag₂₅ is 8, we speculated that Ag₂₆ could be a metastable state. However, we were unable to capture any signals related to Ag₂₅ species in a further ripening process, even when Ag₂₆ was nearly degraded (See Fig. R4 below). Additionally, we noticed that positively charged PPh₄⁺ was introduced in the reaction when a mixed solvent of methanol and dichloromethane was used, as reported for the synthesis of Ag₂₅ (*J. Am. Chem. Soc.* **2015**, *137*, 11578) or its alloyed analogs (*J. Am. Chem. Soc.* **2015**, *137*, 11880). Thus, we speculated that Ag₂₅ could be synthesized and stabilized in some specific conditions, but not in the conditions used in our experiments.

Figure R3. UV-vis absorption spectrum of (a) the final product using AgSPhMe₂ as precursor, and (b) Ag₄₄ nanoclusters.

Figure R4. ESI-MS spectrum of $\text{Ag}_{26}/\text{Ag}_{44}$ mixture after a long-time ripening.

Comment 4. Overall, experimental is splendid, and phenomena are very interesting to me. But, as I mentioned above, there are not more description about reasons of these interesting phenomena very much. If authors will think about this point, this work will be better.

Response: Thanks for the positive comments. We believe the above descriptions give better insight into the phenomenon we reported.

Comments from Reviewer #2:

General Comments: *This report claims to show that Ag nanoclusters are formed through a evolution process that involves intermediates containing tens of Ag atoms. Electrospray method was used to characterise the intermediates.*

The ESI MS method shows only charged ions in the reported analysis conditions. Neutral species will not be detected.

Some of the authors previously reported (reference 31) a similar evolution study on Au₂₅(SR)₁₈. Judging both from citations and personal observation, the novelty and usefulness of ref. 31 (and the current report) is unclear.

The conclusions are only partly reliable because only negatively charged species are observed. Other literature references for electrospray method, especially Ag clusters, should be cited. The report claims to provide "valuable new insights". This referee does not see the interest of this work to others in the community.

Response:

1) Regarding ‘only negatively charged species are observed’

Firstly, we should emphasize that we have tried to capture intermediate species in both positive and negative mode of ESI-MS, as we already indicated in Methods section and Supplementary Fig. 4. In the positive mode, no peaks with Ag isotope patterns could be observed throughout the reaction, thus we focused our analysis on the spectra obtained in negative mode.

Secondly, it is reasonable to assume that neutral species will be unstable in our system. The solvents used in our reaction are methanol and acetonitrile, and the neutral Ag-thiolate complexes are insoluble in such solvent system (e.g. the light yellow precipitate). As the intermediates are formed by NaBH₄ reduction of a Ag-thiolate complex, a neutral species should contain less thiolate ligands due to the accumulation of Ag(0), which would further reduce its solubility. Additionally, neither the coulombic repulsion between particles nor solvation of polar solvents used in our reaction will support the stability of neutral species.

Thirdly, it’s reasonable that all of the intermediate species are negatively charged. The influence of solubility in the mixed solution of methanol and acetonitrile has been explained above. Additionally, the NaBH₄ reducing agent provides an abundance of electrons during the reaction. It’s therefore reasonable for the intermediate species to carry negative charges by combining with these extra electrons.

2) Regarding the ‘valuable new insight’ and ‘the novelty and usefulness’ of our paper

Firstly, the emphasis of our study was identification of the intermediate species formed during the evolution of thiolate-stabilized Ag nanoclusters and to systematically explore the evolution mechanism. To the best of our knowledge, the current study is the first successful work to achieve both goals. Most interestingly, the evolution mechanism of thiolate-stabilized Ag nanoclusters we have identified in the present study is distinct from the traditional “bottom-up” evolution mechanism reported in thiolate-stabilized Au nanoclusters (*J. Am. Chem. Soc.* **2014**, *136*, 10577). This is a very exciting and novel finding that challenges existing assumptions and misconceptions that Ag and Au nanoclusters are similar in their formation mechanism. According, we expect the results of our paper to be of broad scientific interest.

Secondly, our research demonstrates that it’s possible to obtain several kinds of stable Ag nanoclusters with a single kind of thiolate capping ligand, whilst the primary literature generally suggests that only a single kind of atomically precise Ag nanoclusters can be synthesized for a particular capping ligand (*Angew. Chem. Int. Ed.* **2009** *48*, 5921; *J. Am. Chem. Soc.* **2015**, *137*, 11550; *J. Am. Chem. Soc.* **2015**, *137*, 11880). Furthermore, we find that the size (number of Ag atoms) of intermediate Ag-thiolate clusters formed during the top-down evolution plays critical role in the reaction routes to different stable final products. In theory, this gives us a route to control final product size by rationally designed reaction parameters, and will therefore guide future synthetic efforts towards new thiolate-stabilized Ag nanoclusters

Summarizing the aforementioned novelties, our work pinpoints and addresses several critical problems in the synthesis of atomically precise Ag nanoclusters, and paves way for the discovery of new atomically precise Ag nanoclusters in the near future.

Comments from Reviewer #3:

General Comments:

Summary of Key Points

ESI-MS was used to monitor reactive intermediates and discretely sized Ag nanocluster products over time using two bulky, aromatic thiols as capping agents. The ESI-MS spectra provided here does not show signals corresponding to low molecular weight Ag-thiolate monomers or oligomers. This indicates that the Ag17 and Ag44 product nanoclusters in this work are not formed through a “bottom-up” approach, which is commonly reported for Au clusters; this is an original, significant and surprising result. The reactive intermediates instead appear to be clusters themselves, which are similar in atom count to the final nanocluster product, and are likely “size-focused” to the observed discretely sized cluster products. Based on this, the authors propose relative amounts of ligand:metal:charge (denoted M) that predisposed each clusters formation based on shell closure magic numbers and suggest that larger Ag clusters could be formed if the correct “M ratio” is observed in the intermediate cluster.

Validity

There are no major flaws in the work presented here to indicate that it is unfit for publication.

Originality and Significance

The absence of oligomeric precursors in the formation of discrete Ag clusters is an original and significant result because it indicates that these Ag nanoclusters are formed via a different pathway than similarly sized Au clusters. The “M ratio” provided here could be useful to predict relative ratios of precursor chemicals to force the formation of larger clusters with discrete sizes offering some semblance of size control. This could be of broad scientific interest.

Response: We thank the reviewer for his/her very positive comments.

Comment 1.

Data and Methods

In general, the science of the paper is sound, but there were a handful of places where I wanted a few more details to either lend support or to be fully convinced of their claims. For example, the cluster synthesis appears to be relatively standard and both UV-Vis and TEM supports the formation of the Ag44 as seen in the literature. However, for the sake of reproducibility, they should indicate the amount of time it

takes for the yellow precipitate precursor to form before reduction to clusters. Since this is also an ‘intermediate’ in the synthesis of their clusters, I’m curious as to its stability/shelf life and whether differently sized clusters are seen when this is aged for hours or days? Is there any ESI-MS data on this precursor precipitate to get an idea of how “finite” these precursor chains are (i.e. is it the same atom count as the proposed intermediate clusters?) This could lend support to their proposed mechanistic pathways.

Response: Thanks for the insightful suggestions. The reaction of thiol with Ag^+ ions in solution was very fast. The yellow precipitate was formed immediately after mixing Ag^+ with thiol. Though not precisely controlled in all reactions, NaBH_4 was typically added to this suspension to start the reduction reaction within several minutes of Ag-thiolate complex formation (please see section ‘Synthesis of Ag nanoclusters’ in main text and Methods). No obvious difference was observed between repeat batches of reactions in our experiments. As suggested, we also assessed the stability of this yellow precipitate, and conducted the same reaction after storing it for 36 h. As shown in Fig. R5, similar final products were obtained based on their UV-vis absorption and ESI-MS spectrum.

Figure R5. (a) UV-vis absorption and (b) ESI-MS spectrum of the final product obtained by reducing Ag-thiolate complex precursor stored for 36 h.

We were unable to perform any ESI-MS measurements on the Ag-thiolate precipitates, as they were insoluble in all common solvents we tested. This property is probably due to the so-called argentophilicity (Ag-Ag interaction), which has been observed in Ag-thiolate complexes (*Eur. J. Inorg. Chem.* **2004**, 2004, 78). This report also provides a possible structure of the Ag-thiolate precipitate, where ‘finite’ Ag-thiolate chains would connect with each other by Ag-Ag argentophilicity interactions to form ‘infinite’ Ag-thiolate precipitates.

Comment 2.

The ESI-MS data clearly indicates the formation of discrete Ag₁₇ and Ag₄₄ product clusters over time. There is also a noted absence of smaller MW peaks that would be attributed to small Ag(I)-thiolate monomers or oligomers, which supports the assertion that a bottom up mechanism may not operate here. Instead the reactive intermediates observed with ESI-MS appear to have a very broad range that centers around the peak observed for each cluster product. The broad intermediate peaks decrease over time as the product cluster signals intensify, indicating that size-focusing must be occurring on both clusters as the reaction proceeds to completion. The authors note that Ag₁₇ could be formed from “intermediate clusters” of 22 – 34 atoms and Ag₄₄ from 35 – 57 atoms and provide a formula based on the amounts of metal, ligand and charge to predict which cluster forms under which conditions. It appears both clusters do form in a unique way based on the identity and timing of ESI peaks in Figure 3. However, I am not entirely convinced that the “intermediate clusters” giving rise to the ~2800 m/z peaks aren’t also being used as a scaffold for further reduction into the larger Ag₄₄ cluster in addition to or instead of forming Ag₁₇. How can they be sure these higher MW intermediates form only Ag₁₇ and not also Ag₄₄? Is there a ratio of intensity for the appearance/disappearance of the useful peaks to completely support this assertion? It appears to me that there are many peaks around the Ag₁₇ cluster that disappear over time as the Ag₁₇ peak becomes more intense, which could be the only intermediates for Ag₁₇. And there is a huge gap in the ESI spectrum between the Ag₁₇ peak and the purported intermediate clusters that give rise to Ag₁₇. While for Ag₄₄, it appears like an early ~2800 m/z intermediate that could undergo Ostwald ripening to form larger intermediate clusters that are later size focused into the major Ag₄₄ product. A quantifiable metric would be of help to rule this possibility out.

Response: Thanks for the insightful comments and suggestions.

Firstly, we should point out that the peaks around the Ag₁₇ cluster are not intermediates as we couldn’t observe any characteristic isotopic peaks of Ag in them. Actually, as we indicated in our manuscript and Supplementary Fig. 4, they correspond to species related to Na⁺ and NO₃⁻. Their intensities decreased as the reaction proceeded.

Secondly, based on the above fact, the formation of Ag₁₇ should be related to the intermediate Ag-thiolate clusters which show ESI-MS signals in the range of m/z 2000-5000. As shown in Fig. R6, we followed the attenuation of six intermediate Ag-thiolate clusters peaks to show their direct relationship with the formation of Ag₁₇ (Positions of these peaks are also shown in Supplementary Fig. 7 and Supplementary

Table 1). Obviously, there is good agreement between the disappearance of the intermediates and appearance of Ag₁₇ (Please also see Supplementary Fig. 8).

Figure R6. Variation of the intensities of intermediate species in the formation process of Ag₁₇. Intensity of each species is calculated by the signal-to-noise ratio of the maximum of each series of peaks. Calculation of the intensity of Ag₁₇ is based on Ag₁₇(SPh-*t*Bu)₁₂³⁻. Note that the peak is nearly invisible when its signal-to-noise ratio is less than 2.

On the contrary, Ag₄₄ starts to evolve after a reaction time of 4 min (Please see Fig. 3 in Main Text and the corresponding peaks intensities collected in Fig. R7 below). At this stage of reaction, six representative intermediates we have analyzed were almost consumed. Thus, we can conclude that these species giving rise to *m/z* ~2800 peaks have a close relationship with the formation of Ag₁₇ rather of Ag₄₄.

Figure R7. Variation of the intensities of representative intermediate species in the formation process of Ag₄₄. Intensity of Ag₄₄ is calculated based on its main fragment in ESI-MS: Ag₄₃(SPh-*t*Bu)₂₈³⁻. These intermediate species were almost consumed when the Ag₄₄ started to evolve after a reaction time of 4 min.

Comment 3.

Given that the kinetics of cluster formation were too rapid in ESI to be of practical use for detecting intermediates, they reduced the volume of methanol by half and added 50 mol % of NaOH to slow the reaction progress down. I presume this addition of hydroxide is to quench the active hydrogen, but the authors' do not explicitly address the role of hydroxide ion and how it is involved in the mechanism they propose. They do note that the products formed in the presence of hydroxide ion were less stable over time, and showed evidence of other un-explained fragments in ESI, so how can they be sure the same mechanism will operate in neutral conditions versus basic conditions? Was the pH explicitly monitored in both synthetic conditions? Also, did the addition of hydroxide cause any Ag(OH) to precipitate out? A gravimetric type of analysis here could indicate how much Ag is "tied up" in the precipitate cluster and how much is free in solution, which could be useful for a more accurate determination of its elemental composition. Additionally, if the hydroxide does in effect lower the amount of hydride in solution, did the authors try the experiment using less borohydride to see if the same ESI peaks are observed in each condition? In my experience, the amount and/or "freshness" of this reducing agent can noticeably change the size distribution of product clusters.

Response: Thanks for the insightful comments. Given the fact that borohydride needs to interact with H^+ to fully release its reducing capability, sodium borohydride is often used with ice cold water or some other solvents that carries -OH. We introduced OH^- to the solvent by adding excess NaOH to suppress the concentration of H^+ , and to slow down the reduction kinetics of borohydride. In the reaction without NaOH, the solution pH was measured to be 9.3, while by the addition of NaOH, the pH of the mixture was adjusted to above 14 (out of range of the pH meter). (Please see the corresponding descriptions in section 'Capture of reaction intermediates by ESI-MS' in the main text.)

As we have also captured the weak signals of the intermediate species when NaOH was not introduced (Supplementary Fig. 2), we compared them with the signals we obtained in Fig. 3a to evaluate whether the intermediates (mechanism) were changed after the addition of NaOH. As shown in Fig. R8 below, the intermediate species obtained in the system without NaOH (top panel) have a perfect coincidence with the intermediate species obtained with NaOH addition, as shown in Fig. 3a (bottom panel) in the range of m/z 2200-2800 by judging 32 species. (Please see Supplementary Fig. 3)

Figure R8. Comparison of the intermediate species captured in two different conditions.

The reduction reaction started as soon as the NaOH was added, since methanol was added at the same time. We didn't observe any AgOH species in ESI-MS. The reaction solution kept clear throughout the reaction, and we didn't observe any precipitate by the centrifugation of the reaction solution at 12000 rpm. Taken together, we can conclude that there is no AgOH or Ag₂O precipitation during the reaction.

We also tried to detect Ag⁺ in solution after the removal of Ag-thiolate precipitate by centrifugation. Cl⁻ was introduced through a water/methanol mixed solution of NaCl (water/methanol = 1/50). As shown in Fig. R9, no AgCl precipitate could be found.

Figure R9. Detection of free Ag⁺ in solution.

Additionally, we have tried to use a different amount of borohydride in the experiment and observed similar ESI-MS results. When 7 mg of sodium borohydride was used

(and NaOH was added at the same time), we detected similar ESI-MS signals corresponding to Ag-thiolate intermediates during the reduction process (Fig. R10a). The final products are still Ag₁₇ and Ag₄₄ (Fig. R10b), while Ag₄₄ is the main product suggested by its UV-vis absorption spectrum (Fig. R10c).

Figure R10. Reaction intermediates and final products using a different amount of sodium borohydride.

Comment 4.

After a period of hours, ESI indicates the clusters had degraded into small monomers and oligomers (typical as preliminary intermediates for Au clusters). These were used as a control to see if a bottom up approach could work for the formation of the discrete clusters by adding NaBH₄. When they did this, AgNPs formed, but they were much larger than Ag₁₇ and Ag₄₄, which supports that maybe a bottom up mechanism doesn't work for these small 'electronically stabilized' clusters, but perhaps could work for the larger 'geometrically stabilized' clusters. However, it seems that the authors added half as much borohydride for this control reaction than for the initial reaction – there is no indication of the sample volume in this control reaction, so these may be consistent molar ratios with the earlier experiment, but without more details it's hard to know for sure. As stated earlier, the relative amount of borohydride used to generate noble metal nanoparticles has an effect on the size of the clusters generated, so this should be addressed to ensure it's a proper control. This is analogous to the

earlier point, where the amount of “active” borohydride should be less with the addition of hydroxide ion. Does this change in the amount of reducing agent affect the size of these clusters?

Response: Thanks for the insightful comments and suggestions. Sorry, we made a mistake here by choosing the wrong conditions for a control experiment. The influence of the amount of borohydride added in the reaction has explained above in the response of **Comment 3**. Here, we repeated the ‘bottom-up’ process by adding the same amount of borohydride (10 mg) to achieve a proper control experiment. After full degradation of the Ag nanoclusters, 10 mg of fresh sodium borohydride was added and the UV-vis absorption spectrum and TEM images of the final product were included in Fig. R11 and Supplementary Fig. 13. As shown in Figure R11b, Ag nanospheres with particle size above 3 nm could be observed. This data could explain the broadening and slight red-shift of the plasmonic peak of Ag nanoparticles. (Please see section ‘Control experiment through a bottom-up process’ in Main Text and Supplementary Fig. 13)

Figure R11. (a) UV-vis absorption spectrum and (b) TEM image of the final product obtained in the control experiment.

Comment 5.

Conclusions

It's reasonable to conclude that the Ag17 clusters synthesized in this work are generated from other similarly sized clusters through size-focusing events that occur shortly after reduction occurs, instead of from smaller oligomeric units that “build up” to the magic number of atoms for shell closure. It is possible the same type of phenomenon is occurring for Ag44, but without a quantifiable metric of relative peak intensities, I'm not entirely convinced that Ag44 isn't formed through smaller clusters

and later Ostwald ripening. Either way, it is a very interesting result/absence of result that no monomers or small oligomers of silver-thiolate complexes are observed as is seen with gold nanoclusters. This does represent a new pathway for cluster formation and could potentially allow researchers to attempt size control of these clusters using specified ratios of chemical precursors.

Response: Thank you for the very positive comments and suggestions.

We have explained the assignment of the intermediates located at ~2800 in ESI-MS by a quantifiable metric (see response to *Comment 2*). We believe the assignment of these intermediates is reasonable. However, we couldn't rule out that Ag₄₄ would not be formed from smaller clusters. Actually, the intermediate Ag-thiolate clusters of Ag₄₄ contain 35-57 Ag atoms. Some of them do have less Ag atoms than the final product (35-43), and we could define them as 'smaller clusters'. The ripening process should exist in the latter reaction, but this ripening process is not the same as Ostwald ripening generally observed in nanoparticle systems, and thus would not contradict the mechanism we proposed. We can distinguish them in following two aspects.

- 1) Our ripening process is accompanied by chemical reduction of the intermediate Ag-thiolate clusters. In traditional Ostwald ripening, the ripening process starts after reduction reactions have preceded to completion.
- 2) The reaction time of our ripening is substantially shorter than the traditional Ostwald ripening. The intermediate Ag-thiolate clusters are metastable and need to be further reduced to give Ag₄₄ nanoclusters. During the reduction, they will lose thiolate ligands. As such, they may have high structural flexibility and their intercluster conversion process (ripening) would occur easily. As a result, the ripening process accompanied by the reduction could be completed within 15 min at room temperature, as indicated in our manuscript. Conversely, the starting material of the traditional Ostwald ripening is the fully reduced product, which would already possess a high degree of stability (although they are less stable than the final product). Thus, its size-focusing process would consume more time and require higher temperatures, as it has been demonstrated in Au₂₅ (*J. Am. Chem. Soc.* **2014**, *136*, 10577; 72 h, 25 °C), Au₃₈ (*ACS Nano* **2009**, *3*, 3795; 40 h, 80 °C) and Au₁₄₄ (*Nano Lett.* **2009**, *9*, 4083; 24 h, 80 °C) nanoclusters.

Comment 6.

Improvements

Generally, the document was written well, but there were a few things that confused me or that I thought could be improved to clarify and strengthen the paper.

- *Adding those missing details or rationalizations stated above would support their assertions and leave less room for misinterpretation.*

Response: Thanks for the suggestions. We have provided responses above in Comment 1-5.

- *The mechanistic description needs to be very clear. The word ‘clusters’ is used for both a product and an intermediate, and the words ‘precursors’ is used for two different types of precursor chains (infinite and finite). It’s easy to get lost in this part of the text in particular. For example, it may be helpful for the reader if the authors explicitly use “intermediate clusters” or even an acronym like IC throughout the text to ensure clarity.*

Response: Thanks for the insightful suggestion. We have carefully considered all the expressions concerning ‘intermediate Ag-thiolate clusters’, and we hope this could help to avoid misunderstandings of our articulation.

- *Some things appear contradictory. For example, they comment on the remarkable stability of Ag₁₇ and Ag₄₄ clusters, but later go on to describe how these particular Ag₁₇ and Ag₄₄ clusters weren’t particularly stable and decomposed within hours. Addressing this with an explanation (that also ties into the mechanism) would be appreciated.*

Response: Thanks for the suggestion. We have provided a detailed explanation in Supplementary Note 1 concerning the stability of Ag₄₄ and Ag₁₇.

- *A little bit of organizational clean up could improve the readability. There are a few awkward sentences, and odd conclusions drawn at random points in the text that either don’t follow the previous statement, belong elsewhere or are simply redundant.*
 - o *Line 23 spacing typo*

Response: As suggested, we have addressed this issue.

- o *line 62 first sentence seems out of place/awkward with rest of text*
- o *line 64 weird conclusion from previous statement*

Response: These two points have close relationship, so respond to both together. Before the present study, the bulky ligand SPh-*t*Bu was used to stabilize Ag₁₇ rather than Ag₄₄ (in *J. Am. Chem. Soc.* **2015**, *137*, 11550, only Ag₁₇ was synthesized using such ligand). Here, we want to emphasize that the intermediate Ag-thiolate clusters play a critical role in leading to different final products with the same capping ligand. We have articulated this in a better way in the revised manuscript to address this issue.

- o *lines 77-78 redundant from previous.*

Response: Line 77-78 has been removed as suggested.

o Lines 88-90 awkward way to describe the reaction steps.

Response: Language has been modified as suggested.

o 115-117 awkward

Response: Language has been modified as suggested.

o 153 sentence about TEM needs context.

o Line 153 spacing typo

Response: It seems that we provided unnecessary information here. We have removed this sentence to avoid misunderstandings.

o Line 201 missing “of”?

Response: Thanks for the suggestion. The “of” has been added as suggested.

o Lines 220-221 “stable thiolate-stabilized” is redundant

Response: We have removed these words accordingly.

Reviewers' comments:

Reviewer #1 (Remarks to the Author):

The manuscript was revised in respond to my comments and improved to the level of the publication. I support the publication.

Reviewer #2 (Remarks to the Author):

Two reviewers, myself and reviewer 3, have concerns about the intermediates and if they really tell us about the growth mechanism. Proposing that it is 'reasonable' to assume the intermediates are negatively charged does not mean that other positively charged and neutral species are not present, and that a whole growth mechanism story is developed without these ions.

My main concern is the overall usefulness of this work to the community/OTHERS. The authors comment that their previous work, *J. Am. Chem. Soc.* 2014, 136, 10577, was a "very exciting and novel finding". 20% of the citations were self-citations. I can see that you find it exciting and novel.

The abstract reads, "This mechanism differs markedly from the long-established bottom-up evolution process, providing valuable new insights into the synthesis of metal nanoclusters".

Can you explain how your previous work has provided valuable insights into the synthesis of Au metal nanoclusters? Can you show it has influenced OTHER researchers to improve the synthesis? Can you give evidence from literature that it is of broad interest, and useful to others?

Was Ref. 28 (*J. Am. Chem. Soc.* 137, 11880-11883 (2015)) newly added during the revision, highlighted in yellow in the revised document? If this is the case, this is another key reference that has Ag17 already published that is key species in this work. Many (more than a few) of the authors publications are repacking of others published work and built on other works, without giving proper credit to the original contribution.

My opinion about this manuscript has not changed.

I do not recommend publication in *Nature Communications*, but a specialized journal would be appropriate.

Reviewer #3 (Remarks to the Author):

The authors' responses and changes have adequately addressed my previous concerns/points of confusion in the article. Great work, all!

Reply to Reviewers' Comments

Comments from Reviewer #2:

Comment 1. Two reviewers, myself and reviewer 3, have concerns about the intermediates and if they really tell us about the growth mechanism. Proposing that it is 'reasonable' to assume the intermediates are negatively charged does not mean that other positively charged and neutral species are not present, and that a whole growth mechanism story is developed without these ions.

Response: We appreciate the reviewer's thoroughness in reviewing this paper. As indicated in our last response, we also tried to capture positively charged species in positive-mode ESI-MS. As shown in Figure R1a, except for a series of species related to Na^+ and NO_3^- in the low molecular weight region (Supplementary Fig. 4b), no intermediate species were detected in the range of m/z 2000-7000. (Please also refer to Supplementary Fig. 5a for more details)

To further support our assertion, we have also ruled out possible neutral intermediate species in our reactions. $\text{Cs}(\text{OOCCH}_3)$ is widely used to facilitate the ionization of charge neutral nanoclusters by forming Cs^+ adducts, making them amenable to ESI-MS measurement. (*J. Am. Chem. Soc.* **2007**, *129*, 16209-16215; *J. Am. Chem. Soc.* **2009**, *131*, 7220-7221; *Angew. Chem.* **2012**, *124*, 13291-13295; *J. Am. Chem. Soc.* **2014**, *136*, 3673-3679) We therefore added $\text{Cs}(\text{OOCCH}_3)$ into the cluster solution prior to the ESI-MS measurements. As shown in Figure R1b, except for a series of species attributed to Cs^+ and CH_3COO^- in the low molecular weight region (see Figure R1c for detailed analysis), no intermediate species could be detected in the range of m/z 2000-7000. (Please also refer to Supplementary Fig. 5b and 5c)

Based on the above results, we can conclude that positively charged and neutral intermediates were not present as reaction intermediates in our syntheses. (Please refer to Supplementary Fig. 5 and the section 'Analysis of ESI-MS spectra' in the main text)

Figure R1. (a) Positive-mode ESI-MS spectrum of the intermediate reaction mixture in the range of m/z 500-7000. The spectrum shows a series of peaks associated with inorganic ions such as Na^+ and NO_3^- in the reaction mixture (Supplementary Fig. 4b). The inset shows an expanded view of spectrum in the range of m/z 2000-7000. No obvious signals were observed. (b) Positive-mode ESI-MS spectrum of the intermediate reaction mixture in the range of m/z 500-7000. 50 mM CH_3COOCs was added to facilitate ionization of neutral species. The inset shows an expanded view of spectrum in the range of m/z 2000-7000. Except for a series of species related to Cs^+ and CH_3COO^- in the low molecular weight region, no intermediate species could be detected in the range of m/z 2000-7000. (c) Expanded view of spectrum obtained after the addition of CH_3COOCs in the range of m/z 1500-3000. Series of peaks were regularly spaced by 192 Da, corresponding to the molecular weight of CH_3COOCs . The main peaks could be assigned to species with multiple CH_3COOCs and an extra Cs^+ to give positive charge.

Comment 2 and 3. My main concern is the overall usefulness of this work to the community/OTHERS. The authors comment that their previous work, *J. Am. Chem. Soc.* 2014, 136, 10577, was a "very exciting and novel finding". 20% of the citations were self-citations. I can see that you find it exciting and novel.

The abstract reads, "This mechanism differs markedly from the long-established bottom-up evolution process, providing valuable new insights into the synthesis of metal nanoclusters". Can you explain how your previous work has provided valuable insights into the synthesis of Au metal nanoclusters? Can you show it has influenced OTHER researchers to improve the synthesis? Can you give evidence from literature that it is of broad interest, and useful to others?

Response: Since both of these two comments are related to the queries about our previous work, *J. Am. Chem. Soc.* 2014, 136, 10577, we will discuss the two comments above together.

First of all, we don't agree with the reviewer's comment. The paper he/she mentioned has received more than 100 citations (from web of science) over the past three years, and among them more than 77 citations are from other groups. Following this publication, our group has done extensive work on metal nanoclusters, citing this paper as it was related. We believe the citation metrics for the paper suggests that the as-mentioned paper (and similarly the current work in the near future) have generated interests in atomically precise thiolate-stabilized metal clusters in the research community. Indeed, a number of papers have highlighted the findings of our earlier paper, and we have included several examples here:

For example, in *Angew. Chem. Int. Ed.* 2016, 55, 2980-2993, Angelis and co-workers highlighted our work in Figure 13 and said "The potential of ESI-MS for monitoring reactions also expanded to the area of inorganic reaction mechanisms. Particularly, in 2014, the intriguing formation process of thiolated gold nanoclusters (NCs) was addressed by ESI-MS. The fascinating properties of such NCs might be related to their growth mechanism. In the ESI-MS timeresolved study by Jiang, Xie, and co-workers, the gold(I) precursor and the gold NC intermediate species were identified and monitored over time. This time course of gold NC formation revealed the existence of an initial kinetically controlled reduction-growth mechanism and subsequent thermodynamically guided intercluster conversion into $[Au_{25}(SR)_{18}]^-$."

In *Angew. Chem. Int. Ed.* 2015, 54, 3145-3149, Zhu and co-workers cited our work to support their searching for small Au nanoclusters. They said "Recently, the mechanism of the formation of $[Au_{25}(SR)_{18}]^-$ nanoclusters has been studied, in which the $[Au^I(SR)]$ complex is rapidly reduced to small Au nanoclusters, which have 2 or 4 free valence electrons (2e or 4e), and such small nanoclusters finally grow to the

larger Au nanoclusters which have 8 free valence electrons. The $[Au_{15}(SR)_{13}]$ and $[Au_{18}(SR)_{14}]$ nanoclusters were found as the smallest gold-thiolate nanoclusters with 2 and 4 free valence electrons, respectively.”

In *J. Am. Chem. Soc.* **2015**, *137*, 15809-15816, Ma and co-workers gave description to our work and said “Xie and co-workers traced the possible intermediates in the synthesis of $Au_{25}(SR)_{18}$ to investigate the nucleation and growth mechanism by using CO as reducing reagent to slow down the reaction. A series of small-sized AuNC intermediates were identified through ESI-MS.”

In *Chem. Mater.* **2016**, *28*, 3292-3297, Bakr and co-workers cited our paper to support their finding of an isoelectronic addition path in the ligand-induced transformation of Ag nanoclusters. They said “the dimerization of $[Ag_{25}(SPhMe_2)_{13}(SPhF)_5]^{2-}$ seems to follow an isoelectronic addition path giving rise to an electronically stable $[Ag_{50}(SPhMe_2)_{26}(SPhF)_9]^{3-}$ intermediate with a close shell configuration $[50 - 35 - (-3) = 18e^-]$.”

In the past 20 years, great advances have been achieved in the synthesis of different kinds of atomically precise metal nanoclusters (e.g., Au and Ag) and their total structure determination. In addition to the synthesis of metal nanoclusters of particular size, fundamental studies of the underlying mechanisms that govern their synthesis is equally important. Our previous work is the first work to demonstrate the detailed process from Au(I)-thiolate complex precursors to atomically precise Au_{25} nanoclusters by monitoring intermediate species with ESI-MS. A two-stage, bottom-up formation and growth process was unambiguously established: a fast stage of reduction-growth mechanism, followed by a slow stage of intercluster conversion and focusing. Based on this understanding of the detailed synthetic chemistry involved in each step, it is possible to analyze the synthetic strategies that are effective in each step, thereby obtaining target Au nanoclusters.

Based on the results of our previous work, researchers have improved their synthesis of metal nanoclusters, especially those with hydrophilic capping ligands (*Nano Research* **2015**, *8*, 1975-1986; *ACS Appl. Mater. Interfaces* **2017**, *9*, 44856-44863; *Phys. Chem. Chem. Phys.* **2017**, *19*, 12085-12093; *New J. Chem.* **2017**, *41*, 5412-5419; *Adv. Healthcare Mater.* **2016**, *5*, 2528-2535). These nanoclusters have wide applications in the area of environmental science and biology.

Our previous research relating to atomically precise Au nanoclusters is of broad interest and useful in a wide range of research areas. In addition to the methodological development shown above, we would also like to include several other aspects below.

Firstly, our previous research revealed the possibility to analyze detailed evolution process of Au nanoclusters by ESI-MS, which expands the potential of ESI-MS to

monitoring inorganic reaction mechanisms. This is very interesting to the research community of mass spectrometry (*Angew. Chem. Int. Ed.* **2016**, *55*, 2980-2993).

Secondly, based on the detailed mechanism we discovered, researchers have sought improved understanding of the evolution mechanisms involved in other synthetic methods of Au, Ag and Cu nanoclusters (*Chem. Mater.* **2016**, *28*, 3292-3297; *Chem. Mater.* **2016**, *28*, 8385-8390; *J. Phys. Chem. C* **2015**, *119*, 9988-9994; *J. Phys. Chem. Lett.* **2015**, *6*, 2976-2986; *Nanoscale* **2017**, *9*, 8240-8248).

Thirdly, we have mapped out the species involved in the evolution process of Au₂₅ nanocluster, providing a series of “magic size” for future synthetic work. (*J. Phys. Chem. C* **2017**, *121*, 10971-10981; *Angew. Chem. Int. Ed.* **2015**, *54*, 3145-3149).

In addition, our result serves as a good guidance in the DFT calculations of Au nanoclusters to predict their structure and structural evolution in the growth (*J. Phys. Chem. C* **2016**, *120*, 13739-13748; *J. Phys. Chem. Lett.* **2015**, *6*, 1390-1395; *J. Am. Chem. Soc.* **2015**, *137*, 15809-15816).

In summary, our previous work relating to the synthesis of Au₂₅ nanoclusters and their formation mechanism has stimulated a lot of research activity, and is one of the definitive reference works in the field. In our current paper, we reveal for the first time the synthesis mechanism of atomically precise thiolate-stabilized Ag nanoclusters, which occurs via a new approach (*i.e.*, a mechanism quite distinct from the bottom-up mechanism determined for Au nanoclusters). This finding will be of great interest to researchers in the field of metal nanoclusters. We anticipate that our current manuscript will also become a “definitive reference work in the field.”

Comment 4. Was Ref. 28 (*J. Am. Chem. Soc.* *137*, 11880-11883 (2015)) newly added during the revision, highlighted in yellow in the revised document? If this is the case, this is another key reference that has Ag₁₇ already published that is key species in this work. Many (more than a few) of the authors publications are repackaging of others published work and built on other works, without giving proper credit to the original contribution.

Response: No, Ref. 28 (*J. Am. Chem. Soc.* **2015**, *137*, 11550-11553) was included in our first submission, and it was not highlighted in yellow in the revised document. Yes, Ag₁₇ has been successfully synthesized in the community. Our manuscript acknowledged this. Ref. 28 was cited and discussed in several places in our manuscript (*e.g.*, Introduction section and Supplementary Fig. 1).

However, as discussed in our last response letter, we should further mention that the novelty of the present work extends far beyond simply showing that we could

synthesize Ag₁₇ and Ag₄₄ nanoclusters. The emphasis of our study was to capture intermediate species formed in the evolution of thiolate-stabilized Ag nanoclusters and to systematically explore the evolution mechanism. We revealed a totally new evolution process leading to the formation of atomically precise thiolate-stabilized Ag nanoclusters. We believe the reviewer should consider this important point.

We believe an insight into the underlying mechanism will provide a better understanding of phenomenon observed in previous studies and facilitate future efforts to synthesize new Ag nanoclusters. Our work is definitely not the repacking of other published works. The reviewer claimed that “*Many (more than a few) of the authors publications are repacking of others published work and built on other works, without giving proper credit to the original contribution.*” This is a very strong claim, and the reviewer might need to provide more detailed information to support this claim before we could respond to this claim.

To summarize, we have carefully considered all points raised by the reviewers during the revisions of our manuscript. The reviewers’ comments have enabled us to polish our manuscript and to provide a more comprehensive story for readers. We believe our responses above should satisfy the remaining queries that the reviewer might have regarding to our manuscript. We also expect that this work will attract great interest from researchers in various fields, including inorganic chemistry, noble metal chemistry, cluster chemistry, mass spectrometry, nanochemistry, and materials science.

REVIEWERS' COMMENTS:

Reviewer #3 (Remarks to the Author):

The additional ESI data provided in the most recent submission indicates that neutral or positively charged intermediates were not observed under the specific experimental conditions used in an attempt to detect them. Perhaps different conditions than these are required to observe such positive or neutral intermediates in ESI if they do exist. For now though, the evidence presented here is convincing and does support the authors' proposed mechanism. I still support this publication and think this addition strengthens the paper.